# A finer mapping of convolutional neural network layers to the visual cortex

**Tom Dupré la Tour**[1]**, Michael Lu**[1]**, Michael Eickenberg**[2]**, Jack Gallant**[1]
[1]UC Berkeley, [2]Flatiron Institute

## Abstract

There is increasing interest in understanding similarities and differences between convolutional neural networks (CNNs) and the visual cortex. A common approach is to use some specific layer of a pre-trained CNN as a source of features to predict brain activity recorded during a visual task. Associating each brain region to the best predicting CNN layer reveals a gradual change over the visual cortex. However, this winner-take-all mapping is non-robust, because consecutive CNN layers are strongly correlated and have similar prediction accuracies. Moreover, this mapping is usually performed on static stimuli, which ignores the temporal component of human vision. When the mapping is performed on video stimuli, the features are extracted frame-by-frame and downsampled using an anti-aliasing low-pass filter, which removes high temporal frequencies that could be informative. To address the first issue and improve the non-robust winner-take-all approach, we propose to fit a joint model on all layers simultaneously. The model is fit with banded ridge regression, where a separate regularization hyperparameter is learned for each layer. By performing a selection over layers, this model effectively removes non-predictive or redundant layers and disentangles the contributions of each layer. We show that using a joint model increases prediction accuracy and leads to finer mappings from CNN layers to the visual cortex. To address the second issue and preserve more high frequency information, we propose to filter the features with a set of band-pass filters. We show that using the envelopes of the filtered signals as additional features further increases prediction accuracy.

## Introduction

Convolutional neural networks (CNNs) were inspired originally by the anatomy of the brain, and they have been remarkably successful in computer vision [1, 2]. However, these networks still fail in many tasks that humans can perform easily. Therefore, there is increasing interest in understanding the similarities and differences between CNNs and the brain. To investigate this issue, a common method is to use some specific layer of a pre-trained CNN as a source of features to fit a brain encoding model [3]. With this approach, many studies have shown that early CNN layers best predict brain activity in low-level visual areas, while late layers best predict brain activity in intermediate and higher-level visual areas, with gradual changes of layer mapping over the cortical surface [4, 5, 6, 7, 8, 9, 10, 11, 12, 13]. A similar approach has also been applied to speech [14, 15] and language tasks [16, 17].

One problem with this approach is that there are strong correlations between CNN activations from one layer to the next. This confound causes different layers to have similar predictive power in encoding models [8, 16, 17, 18]. It is thus hard to separate which part of the predictive power is specific to a layer and which part is shared with other layers. Most studies ignore this issue and select the best-predicting layer for each voxel [5, 6, 8, 9, 14, 19, 11], but this winner-take-all approach is not robust and it ignores potential complementarities between layers. Some studies use variance partitioning [20] or canonical component analysis [21] to disentangle the different layers, but these approaches cannot disentangle more than two or three layers.

3rd Workshop on Shared Visual Representations in Human and Machine Intelligence (SVRHM 2021) of the Neural Information Processing Systems (NeurIPS) conference, Virtual.

To address this issue, we use banded ridge regression, which has been shown to disentangle contributions from correlated feature spaces in encoding models [22]. Specifically, we fit a predictive model using features from all layers at once, grouping the features by layers, and learning optimal regularization for each layer through cross-validation. We show empirically that this joint model performs a selection over layers, effectively removing non-predictive or redundant layers, and disentangling the contributions of each layer on each voxel. Using this joint model increases prediction accuracy, and leads to smoother cortical maps of layer mapping.

A second problem of the conventional approach is that it oversimplifies the temporal aspect of visual processing. Indeed, most studies only use static image stimuli [5, 6, 7, 8, 11, 13], which entirely ignores the temporal component of human vision. Some studies use video stimuli [8, 9] and extract features frame by frame from an image-based CNN. Then, the features are downsampled to the brain imaging sampling frequency (typically 0.5 Hz), using an anti-aliasing low-pass filter [8, 9]. However, this low-pass filter is suboptimal, because it removes valuable high-frequency information contained in the CNN activations. Indeed, a video stimulus induces brain activity linked to movement, and this brain activity has been shown to be poorly predicted by low-pass filtered static features [23].

To address this issue, we first use video stimuli and extract features frame by frame from an image-based CNN. Then, to preserve more high temporal frequency information, we filter the features with a set of band-pass filters, and extract the envelope of the filtered signals. The envelopes are then used as additional features which increase prediction accuracy of the model. Note that we specifically do not use a video-based CNN to be able to compare both approaches on the same CNN architecture. We expect further gain in prediction accuracy by using the set of band-pass filters on features extracted from a video-based CNN.

These two methodological improvements pave the way for high-precision mappings between CNNs and human brains, which may help both designing and interpreting CNNs, and defining high-resolution information pathways over the cortical surface.

# 1 Conventional approach

The conventional approach for mapping CNN layers to brain regions [4, 5, 6, 7, 8, 9, 19, 11, 12, 13] follows the voxelwise encoding model framework [24, 25]. First, brain activity is recorded while subjects perceive a visual stimulus. Then, the same stimulus is presented to a pretrained CNN, activations are extracted from intermediate CNN layers and preprocessed into features (see below). Finally, a regression model is trained on each voxel to predict brain activity from the features.

## 1.1 Feature extraction

To extract features, each image of the stimulus is first presented to a pretrained image-based CNN. Then, the activations of one convolutional or fully-connected layer are extracted (typically after ReLU and max-pooling layers). With a video stimulus, features are extracted frame by frame from an image-based CNN, before being down-sampled to the brain imaging sampling frequency (typically 0.5 Hz). Next, a compressive nonlinearity $x \mapsto \log(1 + x)$ is applied, and features are centered individually along the train set. Then, to account for the delay between the stimulus and the hemodynamic response, features are either convolved with a hemodynamic response function, or duplicated with multiple temporal delays. This process is repeated for each CNN layer.

**Limitations.** With a video stimulus, the feature extraction process contains a down-sampling step to match the brain imaging sampling frequency (typically 0.5 Hz). This down-sampling is typically done with an anti-aliasing low-pass filter [8, 9]. However, this low-pass filter likely removes valuable information from the CNN activations. Indeed, a video stimulus induces brain activity linked to movement, and this activity has been shown to be poorly predicted by low-pass filtered features [23].

## 1.2 Winner-take-all model

In the conventional approach, a separate ridge regression [26] is fit to the features extracted from each layer of the CNN independently. Then, the best layer is selected for each voxel, based on cross-validated prediction accuracy. Finally, differences in terms of selected layers are analyzed across the brain. The approach thus produces a mapping from CNN layers to brain regions.

**Limitations.**   Because of correlation between layers, two layers can lead to very similar predictions accuracies. In this case, the best-layer selection picks one layer over the other, even if the prediction accuracy difference is small. The best-layer selection is thus non-robust. The winner-take-all approach also ignores the possibility that one low-performing layer might contain information that is not captured by the best-performing layer, and which might be relevant for brain prediction. In this case, both layers would be complementary, in the sense that using both of them jointly would lead to better performance than any single one individually.

## 2   Proposed approach

### 2.1   Temporal filtering

To address the limitations of downsampling during feature extraction, we propose to filter the layer activations with eight complex-valued band-pass filters (of frequency bands [0, 0.5], [0.5, 1.5], ..., [6.5, 7.5] Hz). These band-pass filters preserve information from different temporal frequency bands, similarly to spatio-temporal features from [23]. Then, the amplitude of each filtered complex-valued signal is computed to extract the envelope of the signal. The envelope is the slowly varying amplitude modulation of the high-frequency signal. Finally, the envelope is down-sampled (with an anti-aliasing low-pass filter). The features extracted with the eight filters were either used in separate models (one per filter) or in a joint model with all features concatenated.

### 2.2   Joint model

To address the limitations of the winner-take-all model, we propose to use a joint model over all layers simultaneously. In this model, features from all layers are concatenated and a ridge regression is fit on all concatenated features. Fitting a joint model maximizes the model prediction accuracy by taking into account possible complementarities between layers. Then, to map CNN layers to the brain from the joint model, it is necessary to estimate the relative contribution of each layer to the model. To do so, we adapt the product measure [27, 28], a variance decomposition measure that quantifies relative importance at the level of individual features [29] (see Appendix A.1). Our adapted product measure quantifies relative importance at the level of feature spaces. It is thus used to estimate the relative contribution of each layer to the joint model.

**Continuous layer mapping.**   To derive a layer mapping from the joint model, we use a weighted average of layer indices, weighted by variance decomposition ratios. Starting from a variance decomposition $\rho \in \mathbb{R}^m$, where $m$ is the number of layers, negative values are clipped to zero, and $\rho$ is normalized to sum to one. Then, the layer mapping is computed as a weighted average $s = \sum_{i=1}^m i\rho_i$. This formula leads to a continuous layer mapping, which increases both its robustness and its ability to describe gradual changes over the cortical surface.

### 2.3   Banded ridge regression

One issue arises from the proposed approach. Because CNN activations are strongly correlated from one layer to the next, a joint model tends to use almost all layers. Therefore, the continuous layer mapping gives an estimate that is biased toward middle values. To solve this issue, we refine our approach using banded ridge regression [22] instead of ridge regression. Banded ridge regression is a generalization of ridge regression which uses a separate regularization hyperparameter per feature space (see Appendix A.2). All hyperparameters are learned with cross-validation to adapt the regularization strength to each feature space.

By varying regularization strength on each feature space, banded ridge regression is able to select feature spaces that have good predictive power, and ignore those that have little predictive power or that are redundant with other feature spaces. This process leads to a feature-space selection that can improve the generalization performance on the test set. A feature-space selection also reduces the number of features, following the principle of parsimony (Occam's razor). The feature-space selection mechanism of banded ridge regression is similar to the mechanism present in the group lasso [30]. However, contrary to banded ridge regression, the group lasso does not optimize the group scalings (see Appendix A.2). Banded ridge regression is also faster to estimate than the group lasso on large numbers of voxels. In the case of continuous layer mapping, using fewer layers also mitigates the bias toward middle values.

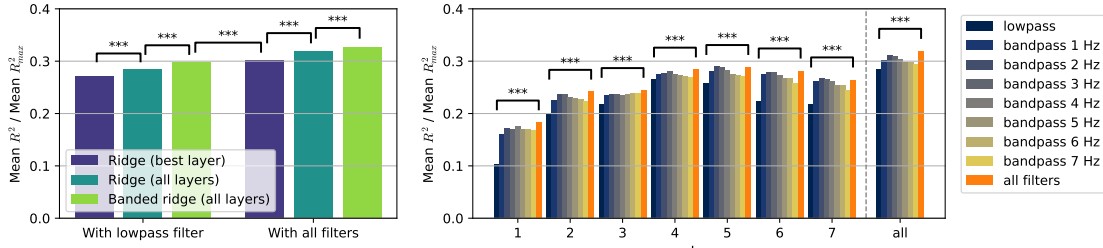

**Figure 1: Model comparison in terms of prediction accuracy.** Prediction accuracy was estimated with the $R^2$ score computed on a test set. The $R^2$ score was averaged over voxels predicted significantly by at least one of the compared models. The average score was normalized by the average noise ceiling $R^2_{max}$. **(Left)** To describe the benefit of the joint model, three models were compared: a winner-take-all model, a joint model fit with ridge regression, and a joint model fit with banded ridge regression. The three models were compared in two settings, using either a low-pass filter, or a set of low-pass and band-pass filters. In both settings, the joint models were better than the winner-take-all model ($p < 10^{-4}$, indicated by ***, paired t-test). Fitting with banded ridge regression was better than fitting with ridge regression ($p < 10^{-4}$). Between both filter settings, the set of filters led to better predictions than the low-pass filter ($p < 10^{-4}$). Overall, the proposed approach increases prediction accuracy from 0.27 to 0.33 (+20%) compared to the conventional approach. **(Right)** To further describe the benefit of each filter, a ridge regression was fit on each combination of filter and layer. For every layer (except layer 4), the low-pass filter led to a lower score than every band-pass filter ($p < 10^{-4}$). Moreover, for every layer, concatenated features from all filters outperformed the low-pass filter features ($p < 10^{-4}$).

## 3 Results

This section presents a series of experiments comparing the proposed approach to the conventional approach. The experiments used functional magnetic resonance imaging (fMRI) brain recordings of a subject watching movie clips [23]. Feature extraction was based on a pretrained image-based CNN "Alexnet" [31] implemented in PyTorch [32]. See Appendix A.3 for more details about model fitting.

### 3.1 Comparing prediction accuracy

To compare the proposed approach with the conventional approach, we fit multiple regression models and estimated their prediction accuracy on a test set using the $R^2$ score. To summarize the performance of each model into a single score, the $R^2$ score was averaged over all voxels significantly predicted by at least one of the compared models ($p < 0.01$, permutation test, see Appendix A.4). The average $R^2$ score was normalized by the average noise ceiling $R^2_{max}$ (see Appendix A.5).

To investigate the benefit of our proposed approach, three models were compared: a winner-take-all model, a joint model fit with ridge regression, and a joint model fit with banded ridge regression. All three models were fit either on features extracted with a low-pass filter, or on features extracted with a set of low-pass and band-pass filters. The results (Figure 1 (left)) first show that the joint model outperformed the winner-take-all model, which demonstrates the benefit of using complementary information from multiple layers. Second, the results show that fitting the joint model with banded ridge regression instead of ridge regression further improved prediction accuracy. It thus demonstrates the benefit of ignoring redundant layers to maximize prediction accuracy. Third, the results show that using a set of filters instead of a low-pass filter further improved prediction accuracy, which demonstrates that valuable information is contained in the high temporal frequencies. Finally, all three improvements (the joint model, banded ridge regression, and the filter set) can be combined to improve the average prediction accuracy from 0.27 to 0.33 (+20%).

To further detail the benefit of using the set of filters, a separate ridge regression was fit on each combination of layer and filter, including concatenated layers and concatenated filters. The results are listed in Figure 1 (right), where each column corresponds to a separate ridge regression fit. The results show that for each layer (except layer 4), each band-pass filter outperformed the low-pass filter. Furthermore, for each layer, concatenating features from all frequency bands (with both band-pass and low-pass filters) outperformed the low-pass filter features. These results are consistent with what is observed with spatio-temporal Gabor features [23].

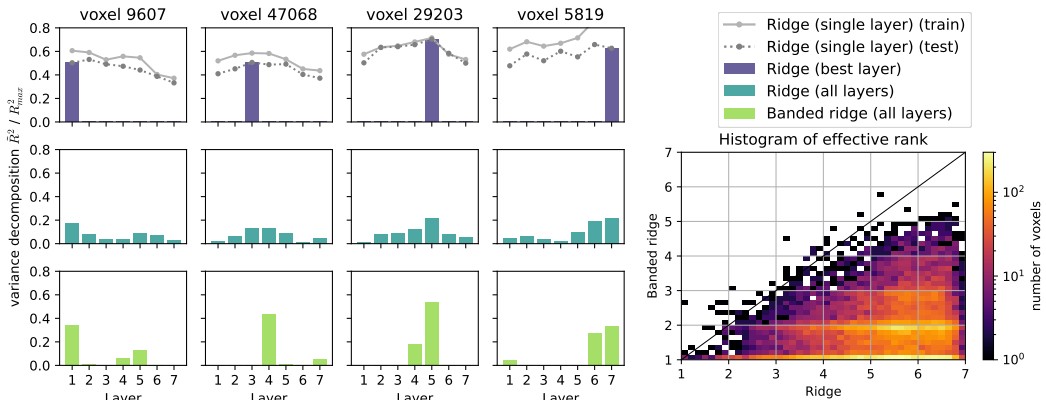

**Figure 2: Model comparison in terms of sparsity at the layer level.** To quantify sparsity at the layer level, we estimated the contribution of each layer to the explained variance ($R^2$ score). The variance decomposition sums up to the $R^2$ score normalized by the noise ceiling $R^2_{max}$. **(Left)** Comparison of three models in four individual voxels. The winner-take-all model (*top*) uses only the best layer, ignoring potential complementarity between layers. The joint model fit with ridge regression (*middle*) uses almost all layers to make the predictions. The joint model fit with banded ridge regression (*bottom*) performs a feature-space selection leading to sparsity at the layer level, and allowing multiple layers to be used simultaneously. **(Right)** To summarize this sparsity profile into a single metric, the effective rank was computed on the variance decomposition. The effective rank estimates the number of layers effectively used in each voxel. The histogram of effective rank over voxels shows that banded ridge regression (*vertical*) led to a sparser model (smaller effective rank) than ridge regression (*horizontal*). This increase in sparsity explains why banded ridge regression led to better prediction accuracy than ridge regression (Figure 1).

## 3.2 Comparing layer sparsity

In the previous subsection, banded ridge regression is shown to outperform ridge regression in terms of prediction accuracy. To explain this difference, we show in this subsection that banded ridge uses fewer layers than ridge regression (*i.e.* it gives rise to a sparser model). This layer sparsity is similar to the one induced by the group lasso [30] (see Appendix A.2). Figure 2 (left) shows examples of variance decomposition over layers of different models. In each voxel shown, the joint model fit with banded ridge regression effectively used fewer layers than when fitting with ridge regression.

To quantify layer sparsity over all significantly predicted voxels ($p < 0.01$, permutation test), we computed the effective rank [33] based on variance decompositions over layers. Starting from a variance decomposition $\rho \in \mathbb{R}^m$, negative values were clipped to zero, and $\rho$ was normalized to sum to one. Then, the effective rank was defined as $\tilde{m} = \exp(-\sum_{i=1}^m \rho_i \log(\rho_i))$. The effective rank is a continuous measure of the number of layers effectively contributing to the model. It is equal to $k$ when the variation is equally split between $k$ layers. Comparing the distribution of effective rank over voxels, Figure 2 (right) shows that banded ridge regression led to a sparser joint model than ridge regression. This sparsity explains why banded ridge regression led to better prediction accuracy than ridge regression.

## 3.3 Comparing layer mapping

The proposed approach not only improves the prediction accuracy compared to the conventional approach, but it also provides a finer mapping of CNN layers to the visual cortex. In the conventional approach, each voxel is associated with the index of the best-predicting layer, leading to a discrete layer mapping. In the proposed approach, a joint model is fit on all layers, and a continuous layer mapping is computed based on the variance decomposition over layers. Figure 3 shows the layer mapping projected on the cortical surface of the visual cortex. Compared to the discrete layer mapping, the continuous layer mapping produced a smoother gradient over the cortical surface, without requiring explicit smoothing or a smoothing prior. The continuous layer mapping built with ridge regression has a reduced range of value, because the model tends to use all layers in each voxel (Figure 2). This reduced range can be mitigated by using banded ridge regression rather than ridge regression, to use only a subset of layers in each voxel.

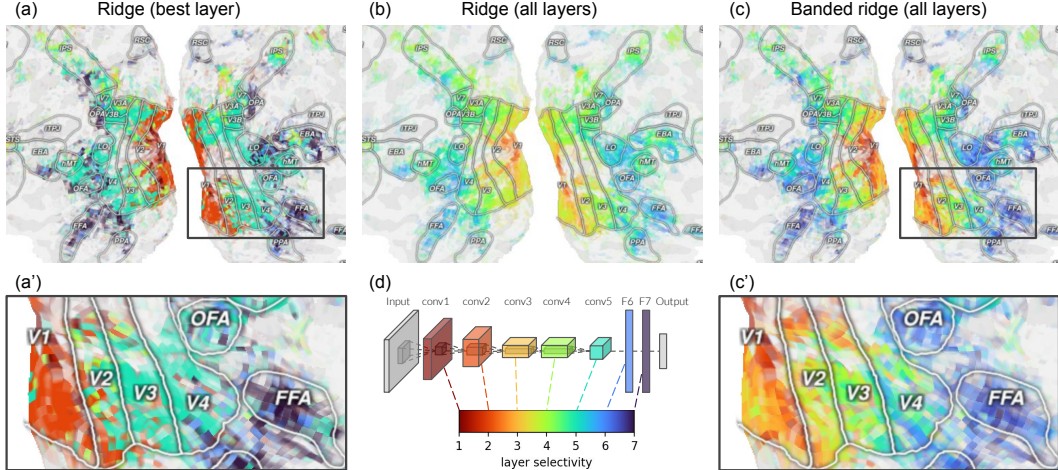

**Figure 3: Model comparison in terms of layer mapping to the visual cortex** The layer mapping of each voxel is computed as a weighted average of the layer indices, weighted by the variance decomposition over layers. For the winner-take-all model, it corresponds to the best-layer index. For visualization, the layer mapping is projected on a flattened cortical surface using Pycortex [34]. **(a)** The winner-take-all model gives a non-robust estimate of layer mapping, because the argmax operator can flip from one layer to another due to small variations in prediction accuracy. **(b)** A joint model fit on all layers gives a continuous measure of layer mapping, with a smooth gradient over the cortical surface without using any smoothing. However, fitting the joint model with ridge regression gives a biased estimate of layer mapping toward middle values, because its variance decomposition tends to use all layers. **(c)** Fitting a joint model with banded ridge regression mitigates this bias thanks to its feature-space selection. The proposed approach led to a finer layer mapping than the conventional approach. **(a', c')** Zoomed views of (a, c). **(d)** Correspondence with the CNN architecture. The layer mapping is derived from the layer indices of a pretrained Alexnet model [31].

## Conclusion

Mapping CNNs to the visual cortex has drawn a lot of attention from both the Machine Learning and the Neuroscience communities. The conventional approach is based on encoding models, which provides a strong framework for model comparison. However, model comparison usually leads to a winner-take-all approach where only the best model is considered, ignoring potential complementarity between models. In this work, we propose to go beyond the winner-take-all approach by fitting joint models over multiple feature spaces simultaneously. The contribution of each feature space to the joint model can then be computed for further interpretation. To fit the joint model, banded ridge regression can be used to induce sparsity at the feature-space level, improving both prediction accuracy and model interpretation. Applied to CNN layer mapping, our approach leads to higher prediction accuracy and to smoother cortical maps. We also highlight the limitation of low-pass filtering when downsampling to the fMRI sampling frequency, and show how to improve it using a set of band-pass filters. Our contributions could be applied to most applications based on the encoding model framework, to improve both model performances and interpretations.

## Acknowledgments and Disclosure of Funding

This work was supported by grants from the National Eye Institute (R01-EY031455), the Office of Naval Research (N00014-20-1-2002), and the Weill Neurohub at University of California, Berkeley. This work was also made possible thanks to the scientific Python ecosystem, including numpy [35], scipy [36], matplotlib [37], scikit-learn [38], pytorch [32], pycortex [34], and himalaya [39].

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

# A Appendix

## A.1 Product measure

The product measure [27, 28] is a variance decomposition measure with many good properties [28, 40, 41, 29]. In particular, the product measure gives a proper decomposition of the variance explained in the training set $\sum_j \rho_j = R^2$. Yet, this property holds only for ordinary least squares and on the training set, because it uses the fact that the error vector $y - \hat{y}$ is orthogonal to the prediction vector $\hat{y}$. We thus adapt the product measure to decompose a definition of the $R^2$ score used for regularized regression method and/or on held-out data, $R^2 = 1 - ||y - \hat{y}||^2/||y||^2$, where $y$ is zero-mean [42]. We then define the variance explained by layer $j$ as

$$\rho_j = \frac{\sum_t \hat{y}_j(2y - \hat{y})}{\sum_t y^2}. \tag{1}$$

Here, $\hat{y}_j$ is the sub-prediction $\hat{y}_j = X_j b_j$, computed on layer $X_j$ alone using the weights $b_j$ of the joint model. Using the definition in (1), the variance explained $R^2$ by the joint model is the sum of the variance explained by all layers: $\sum_j \rho_j = R^2(\hat{y})$. This is an essential property to interpret the measure as a proper variance decomposition. Moreover, when the sub-predictions $\hat{y}_j$ are all orthogonal, the decomposition is identical to using the $R^2$ scores computed on each sub-prediction, $\rho_j = R^2(\hat{y}_j)$. Because in the case of ordinary least squares, the decomposition in (1) is equal to the product measure on the training set, this paper simply refers to it as the product measure.

## A.2 Banded ridge regression

**Definition.** In banded ridge regression, the features are grouped into $m$ feature spaces. A feature space $i$ is formed by a matrix of features $X_i \in \mathbb{R}^{n \times p_i}$, with $n$ samples and $p_i$ features, and each feature space is associated with a different regularization hyperparameter $\lambda_i > 0$. To model brain activity $y \in \mathbb{R}^n$ on a particular voxel, banded ridge regression computes the weights $b_i^* \in \mathbb{R}_i^p$ (concatenated into $b^* \in \mathbb{R}^p$ with $p = \sum_i p_i$) defined as,

$$b^* = \underset{b}{\operatorname{argmin}} ||\sum_i X_i b_i - y||_2^2 + \sum_i \lambda_i ||b_i||_2^2. \tag{2}$$

Similarly to ridge regression, the parameters $\hat{b}_i$ are learned on the training data set, while the hyperparameters $\lambda_i$ are learned by cross-validation [22].

**Related models** Banded ridge regression is similar to the group lasso [30], in the sense that they both lead to a selection of groups of features. However, contrary to banded ridge regression, the group lasso does not optimize the group scalings. In fact, many fixed group scalings have been proposed in the past for the group lasso, using either the square root of the number of features [30], the trace [43] or the empirical rank of the feature kernels [44], or more exotic scalings [45, 46]. Instead, banded ridge regression learns optimal scalings from the data, by maximizing cross-validation performances.

Banded ridge regression can also be formulated as a special case of multiple-kernel learning. Multiple-kernel learning [43, 47] considers generalizations of kernel methods, where instead of a fixed kernel, a collection of kernels is combined to improve the flexibility of the model. Multiple-kernel learning methods traditionally learn the kernel combination jointly with the kernel model [48], yet learning the kernel combination via cross-validation has also been proposed [49, 50, 51]. Banded ridge regression is equivalent to multiple-kernel ridge when learning the kernel combination via cross-validation.

Banded ridge regression can also be linked with automatic relevance determination [52], which learns different hyperparameters to induce sparsity at the feature level. Applying this approach to Bayesian ridge regression leads to sparse Bayesian learning (also known as relevance vector machines) [53, 54], which is the direct Bayesian counterpart of banded ridge regression.

## A.3 Model fitting

**Dataset.** The preprocessed dataset consisted of 73211 cortical voxels recorded every two seconds. Of the 3870 total time samples, 3600 samples were used in the train set (split into 12 runs of 300 samples) and 270 samples were used in the test set.

**Hyperparameter search.** For all models, a leave-one-run-out cross-validation scheme was used to select hyperparameters. For ridge regression, the regularization hyperparameter was selected with a grid-search over 30 log-spaced values ranging from $10^{-5}$ to $10^{20}$. For banded ridge regression, the regularization hyperparameters were selected with a random-search; a thousand normalized hyperparameter candidates were sampled from a Dirichlet distribution, and were scaled by 30 log-spaced values ranging from $10^{-5}$ to $10^{20}$. The best normalized candidate and the best associated scaling were selected on each voxel. Finally, models were refit on the entire training dataset with the selected best hyperparameters.

## A.4 Significance testing

To test significance of the model prediction accuracy on each voxel, a permutation analysis was used. First, brain responses in the training set were permuted by blocks of 5 consecutive time samples. Then, a ridge regression model was fit on the permuted data, and prediction accuracy was computed on the test set. Repeating this procedure 1000 times led to a distribution of prediction accuracy, under the null hypothesis that there is no systematic relationship between model predictions and fMRI responses. Statistical significance was defined as any prediction accuracy that exceeded 99% of all of the permuted predictions ($p = 0.01$), and was calculated separately for each model. Because we estimate statistical significance on multiple voxels, the probability of Type-I errors is largely increased. Therefore, the false discovery rate was used to correct statistical significance levels and control the rate of Type-I error [55]. To be able to compare the average prediction accuracy across models, the average was computed on all voxels significantly predicted by at least one of the compared models.

To test significance of the difference in prediction accuracy between two models, a paired t-test was computed over all voxels significantly predicted by at least on the two models.

## A.5 Noise-ceiling estimation

Because of different sources of noise, a perfect prediction accuracy is impossible to reach. To estimate the upper bound of prediction accuracy achievable in each voxel, the following analysis was used. First, the experiment used in the test set was repeated $q = 10$ times. Then, the average brain response was computed $\tilde{y} = \frac{1}{q} \sum_{i=1}^{q} y_i$, where $y_i$ is the brain response at repeat $i$. All prediction accuracies were computed with respect to the average response $\tilde{y}$. Then, the signal power (also known as the unbiased explainable variance, or noise ceiling) was computed with:

$$P = \frac{1}{q-1}\left(q \operatorname{var}(\tilde{y}) - \frac{1}{q}\sum_{i=1}^{q}\operatorname{var}(y_i)\right), \tag{3}$$

where the temporal variance $\operatorname{var}(y)$ is defined by:

$$\operatorname{var}(y) = \frac{1}{n-1}\sum_{t=1}^{n}\left(y[t] - \frac{1}{n}\sum_{t=1}^{n}y[t]\right)^2. \tag{4}$$

Finally, the maximum achievable $R^2$ score was computed with $R^2_{\max} = \frac{P}{\operatorname{var}(\tilde{y})}$.

In Figure 1, the prediction accuracy is computed with the $R^2$ score with respect to the average response $\tilde{y}$, and averaged over significantly predicted voxels. The average score is then normalized with the maximum achievable score $R^2_{\max}$ averaged over the same voxels. In Figure 2, the prediction accuracy is computed with the $R^2$ score with respect to the average response $\tilde{y}$, divided by the maximum achievable scores $R^2_{\max}$.

