# OpenReview forum: "A finer mapping of convolutional neural network layers to the visual cortex"
_NeurIPS.cc/2021/Workshop/SVRHM — SVRHM 2021 Poster_

### Official Review · Reviewer_e4Ft · 2021-10-21
**Nice methodological improvement**

**Rating:** 8
**Confidence:** 3

**Review:**

The paper deals with prediction of fMRI signals from deep networks. It introduces two variations on previous approaches. First, a band-pass filter bank is used, rather than a traditional low-pass filter, to extract temporal features. Second, rather than predicting each voxel’s activity from a single best network layer, all layers are used, with a regularization parameter chosen independently per layer. Together, these methods substantially improve predictions and produce a smoother map of network-layer selectivity across the cortex.

Pros:
-	The writing is clear and engaging
-	The work is technically well done
-	The approach is successful
-	The work is applicable to an active area of research and likely to be used by others

Cons:
-	I am not sure I fully understood the filter bank. First, the traditional low-pass filter is called an anti-aliasing filter, which makes sense, but is there not also a hemodynamic consideration? Second, the signals are band-pass filtered (e.g. 6.5-7.5Hz) and then low-pass filtered, if I understand correctly, to anti-alias a 0.5Hz sampling frequency. This suggests a low-pass cutoff of 0.25Hz. Combined, these filters are band-pass with a high-pass cutoff much higher than the low-pass one, which is strange. Does this become similar to a less-strange filter and some attenuation? Does it reintroduce aliasing? Does that matter for this application?

Minor:
-	Why does independently varying the regularization make the contributions so much sparser? Any non-uniformity would tend to make the contributions sparser in a sense, but the examples seem very sparse. Are the smaller entries in Figure 2 actually zero?
-	I didn’t know what this meant (line 311): “P-values were corrected for multiple comparisons using the false discovery rate.”

---

> ### Author Response · Authors · 2021-12-07
> **Answer to reviewer e4Ft**
>
> - *“Combined, these filters are band-pass with a high-pass cutoff much higher than the low-pass one, which is strange. Does this become similar to a less-strange filter and some attenuation? Does it reintroduce aliasing? “*
>     - We agree that using a lowpass filter immediately after a bandpass filter would be a poor design. However, our procedure includes an intermediate step between the two filtering steps. After applying the (complex-value) bandpass filter, the envelope of the filtered signal is computed using a complex modulus. Contrary to the filtered signal, the envelope contains low-frequency variations corresponding to amplitude modulations of the high-frequency carrier signal. The envelope can later be low-pass filtered before downsampling to 0.5 Hz. The bandpass filter bank is thus similar to computing a spectrogram of each feature. We revised the paper to clarify this important step.
> - *”Why does independently varying the regularization make the contributions so much sparser? Are the smaller entries in Figure 2 actually zero?”*
>     - Banded ridge regression induces sparsity at the feature-space level. We discuss the link between banded ridge regression and other group-sparsity-inducing models in the appendix. The smaller entries in Figure 2 are not exactly zero because the hyperparameters are not infinite, which lead us to use the effective rank to compute the number of feature space effectively used by the model in each voxel (Figure 2b).
> - *”I didn’t know what this meant: “P-values were corrected for multiple comparisons using the false discovery rate”.”*
>     - This sentence corresponds to a correction of statistical significance levels to account for multiple comparisons. Because we estimate statistical significance on multiple voxels, the probability of Type I errors is largely increased. The false discovery rate is a method to correct statistical significance levels and control the rate of Type I error. We revised the paper to explain the goal of this method. Please see [Benjamini and Hochberg 1995] for more information.

---

### Official Review · Reviewer_ThmT · 2021-10-26
**Interesting alternative approach, though I would appreciate to read more comments about the results and a clearer explanation regarding how the use of videos and temporal filtering fit in the picture.**

**Rating:** 7
**Confidence:** 4

**Review:**

The research presented here aims to investigate the similarities and differences between CNNs and the visual cortex by using a new modelling approach which, instead of relying on a winner take all mapping, fits banded ridge regression starting from all layers simultaneously.
The traditional approach involves fitting one brain encoding model per layer and then calculating for each brain area which layer best predicts brain activity. However, the authors point out that this winner-take-all approach could hide more subtle relations and nuances between the brain and other layers. They therefore propose to fit one single brain encoding model that takes into account all layers simultaneously. They first tried to apply ridge regression, but this caused the results to be biased towards middle values because CNNs’ layers are highly correlated from one to the next. The authors therefore applied banded ridge regression to solve this issue.
The final results show that the application of this alternative approach smooths out the relation between CNN layers and brain areas.

I think the method proposed here is valuable and worth discussing because it is important to constantly monitor the efficacy and validity of the methods we are using.
However, I do have some queries which I’d like the author to address.
* The majority of the paper is related to the regression method used to create the brain encoding model. However, in the methods and the results, the authors suddenly include a paragraph about the limitations of using video stimuli and the inclusion of temporal filtering to overcome this limitation. I would appreciate it if this bit would not come out of the blue but was embedded in the text throughout and if the authors could explain the relation between the temporal filtering and the regression method problems.
* Related to the previous point, a pretrained Alexnet implemented in pytorch was used. However, the traditional Alexnet cannot get movies as input but only static images. How were the movies fed to Alexnet? Frame by frame? Or using a different method?
* The authors claim that the winner-takes-all approach is not good enough because it does not consider the fact that subsequent layers in a CNN are highly correlated. But then, the authors themselves claim that the banded ridge regression is a better method compared to the simple ridge regression because of the correlation between the layers in the CNN. I would appreciate it if they could discuss this further, unpacking the consequences of using one approach or the other.
* Related to the previous point, I would appreciate it if the authors could analyse how similar the results from the traditional method are to the results obtained using banded ridge regression. Though I appreciate the importance of nuances, I would also like to understand the magnitude of the difference because from a purely qualitative look at figure 1 it seems that the traditional method could be not that far off.

---

> ### Author Response · Authors · 2021-12-07
> **Answer to reviewer ThmT**
>
> - *”The authors suddenly include a paragraph about the limitations of using video stimuli and the inclusion of temporal filtering to overcome this limitation. I would appreciate it if this bit would not come out of the blue but was embedded in the text throughout.”*
>     - We agree that the proposed temporal filtering could be better embedded in the text. We have therefore revised the paper so that this aspect is presented in the abstract and the introduction.
> - *”I would appreciate it if the authors could explain the relation between the temporal filtering and the regression method problems.”*
>     - The proposed temporal filtering captures high-frequency information contained in the features that would otherwise be lost when using a lowpass filter. Preserving the high-frequency information is critical to maximize the model performance, but it is not directly linked to the regression method. Temporal filtering is improving the features extraction, independently from the regression method used. We have revised the paper to clarify this relation.
> - *”How were the movies fed to Alexnet? Frame by frame?”*
>     - Yes, movies were fed frame by frame, resulting in features at 15 Hz that have to be downsampled to 0.5 Hz before being used in encoding models. We have revised the methods to make this clear.
> - *”The authors claim that the winner-take-all approach is not good enough because it does not consider the fact that subsequent layers in a CNN are highly correlated. But then, the authors themselves claim that the banded ridge regression is a better method compared to the simple ridge regression because of the correlation between the layers in the CNN. I would appreciate it if they could discuss this further, unpacking the consequences of using one approach or the other.”*
>     - Because of correlations between CNN layers, the feature spaces used in the joint ridge model are very redundant. The joint ridge model is able to leverage complementarity of the feature spaces, but it also draws redundant information from multiple feature spaces, which ultimately leads to a biased layer selectivity (Figure 3). Banded ridge regression can be used to solve this issue. Indeed, banded ridge regression induces group-sparsity, which means that it tends to use only a subset of the feature spaces. Banded ridge regression thus ignores redundant feature spaces (Figure 2), which improves both generalization performance (Figure 1) and interpretation (Figure 3). We have revised the paper to clarify this.
> - *”Related to the previous point, I would appreciate it if the authors could analyse how similar the results from the traditional method are to the results obtained using banded ridge regression. Though I appreciate the importance of nuances, I would also like to understand the magnitude of the difference because from a purely qualitative look at figure 1 it seems that the traditional method could be not that far off.”*
>     - We are sorry that the presentation of our results was not clear. We have revised the text to clarify the following points: The results in Figure 1 show a 20% increase in prediction accuracy ($R^2$ score) between our approach and the conventional approach. Figure 1 also describes the contribution of each of the proposed improvements (bandpass filters, joint model, and banded ridge regression). Additionally, Figure 2 and 3 demonstrate qualitatively how both the joint model and banded ridge regression improve the layer mapping and lead to a smooth gradient of mapping over the cortical surface.

---

### Official Review · Reviewer_q6hm · 2021-10-29
**Nice paper**

**Rating:** 8
**Confidence:** 4

**Review:**

The paper proposes a refined method to study the correspondence of representations between task-trained deep neural nets (DNN; here AlexNet on ImageNet) and neural population activity (here fMRI data across the human visual system). Instead of fitting a separate encoding model for each layer of the DNN and selecting the best matching layer bases on predictive performance of these models, the authors propose to fit a single joint model and perform layer selection by using a banded ridge regression approach, where the regularization parameter for each layer is cross-validated independently. They show that their approach improves over the traditional winner-take-all approach both quantitatively and qualitatively. Overall, the paper is quite well written and easy to follow. I am very supportive of the paper.

One question I would be interested is how the banded ridge regression compares to group sparsity. For group sparsity, the regularization term would be sum_l{sqrt[sum_i(w_il^2)]}, where l is the layer index and w_il the weight of unit i within layer l. This loss encourages sparsity among the groups (here: layers), i.e. tries to use as few layers as possible. It is often used in situations where we expect entire groups of weights to be irrelevant.

---

> ### Author Response · Authors · 2021-12-07
> **Answer to reviewer q6hm**
>
> - *”One question I would be interested in is how the banded ridge regression compares to group sparsity.”*
>     - We agree the comparison between banded ridge regression and the group lasso is interesting. Both methods induce sparsity at the level of group of features, although in a different way. The appendix contains a discussion about the link between banded ridge regression, group lasso, multiple kernel learning, and automatic relevance determination. We have revised the text to better highlight this discussion. Our lab is also currently working on a more in-depth paper about banded ridge regression and its group-sparsity-inducing property.

---

### Official Review · Reviewer_B8hu · 2021-10-31
**an interesting alternative to the "best layer" type of analysis for neural predictions**

**Rating:** 7
**Confidence:** 4

**Review:**

This paper proposes an interesting alternative to the "best layer" type of analysis used for mapping CNN layers onto neural responses. The authors propose using banded ridge regression to jointly fit the feature spaces extracted from all layers of the network. They demonstrate that this method improves the neural predictions over the standard method of only using a single layer for predicting a voxel. The authors additionally improve neural predictions by using a feature extraction method composed of many temporal filters. This type of question is topical for SVRHM as many people are using encoding models to compare neural network responses to those of biological systems, and this paper may create conversation about how some of the underlying decisions in the analysis methods may result in significant differences in neural predictions.

Pros:
1) The comparison between the different types of regression/variance partitioning methods is interesting and provides an alternative to the typical "best layer" approach.
2) The paper demonstrates that the downsampling method used for CNN activations can have a significant influence on how well the model can predict fMRI responses. Determining how to best set up the feature extraction for neural predictions is a crucial step, and the use of multiple bandpass filters is an interesting approach.
3) The method proposed leaves open many questions, which in this case seems positive as it demonstrates that there is future work to be done in this space.

Cons:
1) The analysis seems a bit preliminary and is only tested on one dataset and model. It would strengthen the claims to see the same analysis of banded ridge vs. best layer performed on other datasets (even other types of data that have similar issues, such as neural recordings instead of fMRI).
2) No noise ceiling is given for the fMRI analysis (which would make it easier to determine whether the updated models are maxing out the dataset or if there are still improvements to be made).
3) The feature extraction experiment with bandpass filters is interesting, but it is not mentioned in the abstract or introduction. This seems like an interesting result in itself, and should perhaps be highlighted sooner in the paper (as written, this experiment comes out of nowhere which is a bit confusing for the reader). I also would have appreciated some discussion on how to interpret the usage of the bandpass filters (ie is this a better approximation of the fMRI measurement or do the authors think that it is a better approximation of the underlying neural activity?)

---

> ### Author Response · Authors · 2021-12-07
> **Answer to reviewer B8hu**
>
> - *”1. The analysis seems a bit preliminary and is only tested on one dataset and model.”*
>     - We agree that testing our method on more datasets would be interesting, and we will certainly pursue this idea in the near future. We also hope that our work will inspire other researchers to use and adapt our method on other datasets.
> - *“2. No noise ceiling is given for the fMRI analysis.”*
>     - In the revised paper we have added the noise ceiling to the analysis. The noise ceiling was computed based on ten repeats of the experiment that had been collected in the test set. Figure 1 and Figure 2 were updated to correct prediction accuracy based on the noise ceiling, and a paragraph was added to the appendix explaining the detailed computations.
> - *“3. The feature extraction experiment with bandpass filters is interesting, but it is not mentioned in the abstract or introduction.”*
>     - We agree that the proposed bandpass filter bank is an interesting contribution in itself. We have therefore revised the paper so that this aspect is presented in the abstract and the introduction.
> - *“3. I also would have appreciated some discussion on how to interpret the usage of the bandpass filters.”*
>     - Bandpass filters capture high-frequency information contained in the features that would otherwise be lost when using a lowpass filter. Visual neurons likely respond to high-frequency information and these high-frequency responses are likely reflected in the low-frequency BOLD response measured by fMRI. A similar model improvement can be found in [Nishimoto et al., 2011], where brain activity induced by videos has been shown to be better predicted by spatio-temporal features than by low-pass filtered static spatial features. Preserving the high-frequency information is thus critical to maximize the model performance and to provide an accurate interpretation of the results. We have revised the paper to include this discussion.

---

### Decision · Program_Chairs · 2021-11-02

Accept (Poster)

---

> ### Author Response · Authors · 2021-12-07
> **Thanks**
>
> We thank all the reviewers for their constructive remarks which helped us improve this work. We appreciate the opportunity to answer their questions and clarify multiple aspects of the paper.